# END-TO-END ABNORMALITY DETECTION IN MEDICAL IMAGING

## ABSTRACT

Deep neural networks (DNN) have shown promising performance in computer vision. In medical imaging, encouraging results have been achieved with deep learning for applications such as segmentation, lesion detection and classification. Nearly all of the deep learning based image analysis methods work on reconstructed images, which are obtained from original acquisitions via solving inverse problems (reconstruction). The reconstruction algorithms are designed for human observers, but not necessarily optimized for DNNs which can often observe features that are incomprehensible for human eyes. Hence, it is desirable to train the DNNs directly from the original data which lie in a different domain with the images. In this paper, we proposed an end-to-end DNN for abnormality detection in medical imaging. To align the acquisition with the annotations made by radiologists in the image domain, a DNN was built as the unrolled version of iterative reconstruction algorithms to map the acquisitions to images, and followed by a 3D convolutional neural network (CNN) to detect the abnormality in the reconstructed images. The two networks were trained jointly in order to optimize the entire DNN for the detection task from the original acquisitions. The DNN was implemented for lung nodule detection in low-dose chest computed tomography (CT), where a numerical simulation was done to generate acquisitions from 1,018 chest CT images with radiologists' annotations. The proposed end-to-end DNN demonstrated better sensitivity and accuracy for the task compared to a two-step approach, in which the reconstruction and detection DNNs were trained separately. A significant reduction of false positive rate on suspicious lesions were observed, which is crucial for the known over-diagnosis in low-dose lung CT imaging. The images reconstructed by the proposed end-to-end network also presented enhanced details in the region of interest.

## 1 INTRODUCTION

Deep neural networks (DNN) have shown promising performance in computer vision for various applications such as segmentation, detection and recognition. In medical imaging, DNN-based computer vision is also desirable because that radiologists' routine work requires handling of large amount of data, and the possibility exists that the intensive labor may lead to misdiagnosis (Greenspan et al., 2016; Ying et al., 2016; Li et al., 2017). Furthermore, in some radiation related applications such as computed tomography (CT), low-dose scans are always preferred to decrease the potential harm that ionized radiation may do to human body. The increased noise level in low-dose data made it even more challenging for the radiologists to make correct decisions (Kalra et al., 2004).

Almost all the DNNs in medical image analysis are constructed in the image domain, which is the same domain where radiologists do the observations. However, in most medical imaging modalities, the acquired data are in a different domain from the images, and inverse problems have to be solved to reconstruct the images. For example, magnetic resonance imaging (MRI) acquires data in the Fourier domain, and CT acquires data in the Radon transform domain (Deans, 2007). During the reconstruction, there is great possibility of information lost due to the presence of noise in the measurements, especially for low-dose scans (Pickhardt et al., 2012). To compensate for the noise, iterative methods that exploit prior knowledge about human body have been proposed (Erdogan & Fessler, 1999; Sidky & Pan, 2008; Kim et al., 2017). To achieve better representation of the med-

ical images, DNN based reconstruction methods were also proposed recently (Wu et al., 2017a;b; Gong et al., 2017). However, there is still gap between the objective image quality improvement and its utility in diagnosis, which means that both radiologists and computer aided diagnosis (CAD) systems are working with sub-optimal images.

There is an emerging trend on task-based (end-to-end) signal processing with DNNs in recent years, where the decisions were directly made by the DNNs without explicit intermediate representations. Graves & Jaitly (2014) used DNN for speech recognition directly from audio data without and intermediate phonetic representations. Bojarski et al. (2016) trained a DNN for self-driving cars which learned commands directly from images without recognition of land markers. Iizuka et al. (2016) used a classification criteria for the colorization of grey-scale images. Wang et al. (2011) detected words directly from scenes without doing a two-step text detection and optical character recognition (OCR). It was demonstrated that end-to-end DNNs had improved performance compared to multiple-step learning in these applications.

In this paper, we proposed an end-to-end DNN which predicts the location of abnormalities in the images from the acquisitions. A reconstruction DNN was built first to map the acquired data to the images in order to align the data with annotations made by the radiologists. The DNN approximated a 10-iteration unrolled sequential quadratic surrogates (SQS) algorithm (Erdogan & Fessler, 1999). A 3D convolutional neural network (CNN) was used to detect abnormalities from the reconstructed images. The entire DNN was optimized jointly with regard to the total detection cross entropy loss. The method was implemented on The Lung Image Database Consortium image collection (LIDC-IDRI) from The Cancer Image Archive (TCIA), where we simulated ultra low-dose CT scans from the original patients' data (Armato et al., 2011; Clark et al., 2013; Armato III et al., 2015). The task of the DNN was lung nodule detection, which is essential for early stage cancer screening (Team et al., 2011). The performance of the end-to-end method was evaluated with entropy loss and receiver operating characteristic (ROC), and compared to a two-step approach, where the reconstruction DNN was trained first and the detection DNN was trained on the reconstructed images. Furthermore, the intermediate reconstructed images and features from the end-to-end network were studied for more comprehensive understanding of the DNN.

## 2 METHODOLOGY

In medical imaging, the acquisition domain usually has low coherence with the image domain, which means that a local signal such as a lesion presented in the images will spread out in the acquisition data (Lustig et al., 2008). As the consequence, applying DNN directly on the acquisition data without any prior knowledge requires impractically large receptive field of the network. Furthermore, the data have to be aligned with radiologists' annotations which are in the image domain. Hence, in the proposed end-to-end DNN for abnormality detection, a reconstruction network was applied first to map the data to the image domain, followed by a detection network to detect the lesions.

### 2.1 MATHEMATICAL MODEL OF CT IMAGING

In CT, an X-ray source and detector array are placed on the opposite side of the patient and rotate to detect signals from different angles. The detected data $\mathbf{b}$ are the line integral of the attenuation coefficients inside the patient, which is the image $\mathbf{x}$ to be reconstructed. The reconstruction problem can be written as (Wang et al., 2006):

$$\mathbf{x} = \arg\min_{\mathbf{x}} \|\mathbf{A}\mathbf{x} - \mathbf{b}\|_{\mathbf{w}}^2 + \beta R(\mathbf{x}), \tag{1}$$

where $\mathbf{A}$ is the measurement matrix, $\mathbf{w}$ is the noise weighting matrix which is usually diagonal and related to $\mathbf{b}$, $R(\cdot)$ is the a function that incorporates prior knowledge about $\mathbf{x}$, and $\beta$ is the hyper-parameter to control the balance between data fidelity and prior knowledge terms.

SQS is one of the most widely used optimization algorithms to solve (1), which has simple updating formula and fast convergence rate. The image updating formula can be written as (Erdogan & Fessler, 1999):

$$\mathbf{x}_{n+1} = \mathbf{x}_n - \frac{\mathbf{A}^T \mathbf{w} (\mathbf{A}\mathbf{x}_n - \mathbf{b}) + \beta \nabla R(\mathbf{x}_n)}{\mathbf{A}^T \mathbf{w} \mathbf{A} \mathbf{1} + \beta \texttt{diag}\left(\nabla^2 R(\mathbf{x}_n)\right)}, \tag{2}$$

where $\mathbf{1}$ is an all-ones vector that has the same dimension with $\mathbf{x}$, $\texttt{diag}(\cdot)$ is the diagonal elements of the matrix, and the long division is element-wise operation. Choices of the prior function $R(\mathbf{x})$ include total variation, wavelet transform, etc.(Sidky & Pan, 2008; Chen et al., 2008).

## 2.2 DEEP NEURAL NETWORK FOR IMAGE RECONSTRUCTION

The reconstruction model (1) takes iterative solution and is not compatible with end-to-end DNNs. The hand-crafted prior function $R(\mathbf{x})$ also constrained the expression power of the reconstruction model. One of the solutions is to approximate iterative solutions with finite steps expressed by DNN, and the feasibility has been demonstrated on several algorithms such as ISTA, steepest descend, ADMM, primal-dual, etc. (Gregor & LeCun, 2010; Sun et al., 2016; Hammernik et al., 2017; Adler & Öktem, 2017). By substituting operations on $R(\mathbf{x})$ with neural networks in (2), the updating formula became:

$$\mathbf{x}_{n+1} = \mathbf{x}_n - \frac{\mathbf{A}^T \mathbf{w} (\mathbf{A}\mathbf{x}_n - \mathbf{b}) + f_n(\mathbf{x}_n)}{\mathbf{A}^T \mathbf{w} \mathbf{A} \mathbf{1} + g_n(\mathbf{x}_n)}, \tag{3}$$

where $f_n(\cdot)$ and $g_n(\cdot)$ are CNNs.

Several studies indicated that it is not necessary for the DNN to take the exact form of the algorithm (Adler & Öktem, 2017; Chen et al., 2017; Schlemper et al., 2017). To reduce the computation load, we omitted the $g_n(\mathbf{x})$ on the denominator in the realization. Furthermore, an adaptive step size $\lambda_n$ was added to improve the capacity of DNN. The final formula of the reconstruction network is:

$$\mathbf{x}_{n+1} = \mathbf{x}_n - \lambda_n \frac{\mathbf{A}^T \mathbf{w} (\mathbf{A}\mathbf{x}_n - \mathbf{b}) + f_n(\mathbf{x}_n)}{\mathbf{A}^T \mathbf{w} \mathbf{A} \mathbf{1}}. \tag{4}$$

The initial image $\mathbf{x}_0$ was taken as the results of filtered backprojection (FBP) from data $\mathbf{b}$ with Hann filter. When training the reconstruction network alone, the L2-distance between the reconstructed images from simulated low-dose data and the original ground truth was minimized. The structure of the reconstruction DNN is shown in figure 1. The structure of the sub-CNNs was constructed following the work of Adler & Öktem (2017).

The gradient backpropagation through the reconstruction module

$$\mathbf{y}_n = \mathbf{A}^T \mathbf{w} (\mathbf{A}\mathbf{x}_n - \mathbf{b}) \tag{5}$$

has to be specified for the training. Denote the loss of the network as $l$, the chain rule of derivatives gives that

$$\nabla_{\mathbf{x}_n} l = \mathbf{A}^T \mathbf{w} \mathbf{A} \nabla_{\mathbf{y}_n} l. \tag{6}$$

When constructing the system matrix $\mathbf{A}$, we used multi-slice fan-beam geometry, which is shown in figure 2. The projection data in the multi-slice fan-beam geometry can be approximately acquired by rebinning (resampling) from the helical geometry (Noo et al., 1999), which was adopted by most clinical CT scanners.

With the multi-slice fan-beam geometry, the system matrix $\mathbf{A}$ became the same for each layer of the image, which would greatly reduce the amount of memory needed during the training, since only a few layers from $\mathbf{x}$ was needed to feed the DNN instead of the entire 3D volume. To utilize the correlation between adjacent slices during the reconstruction, the input to the DNN contained $L$ successive layers of images, which composed $L$ channels for the CNNs $f_n(\cdot)$.

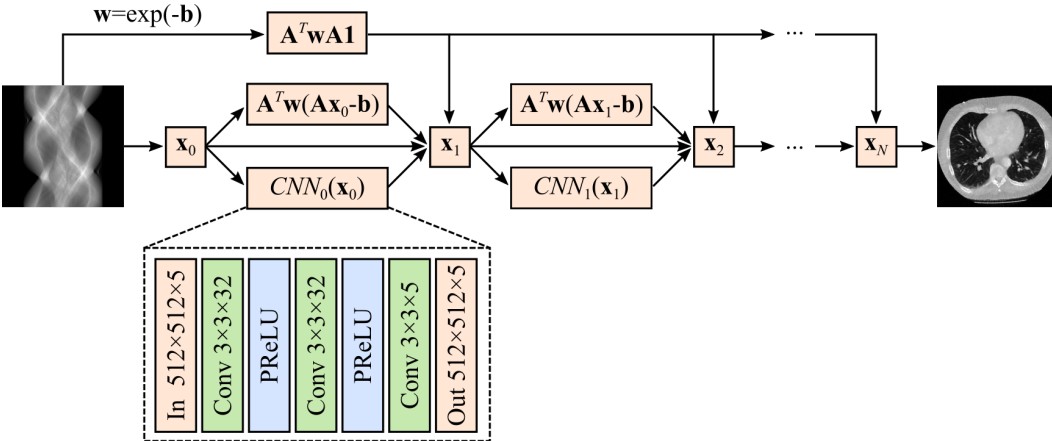

Figure 1: The structure of the reconstruction neural network. $\mathbf{x}_0$ was reconstructed from the projection data by FBP. The convolutional layers used symmetric padding on the image boundary.

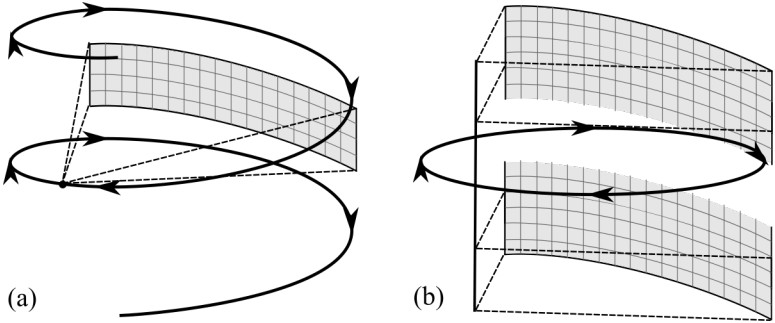

Figure 2: Geometry of the CT imaging system: (a) the helical scanning geometry in most clinical CT systems; (b) the multi-slice fan-beam geometry after projection data rebinning (resampling).

### 2.3 DEEP NEURAL NETWORK FOR LUNG NODULE DETECTION

Pulmonary nodules (lung nodules) are abnormal tissues in lung, some of which may develop into cancer. It is important to detect lung nodules for early stage cancer screening (MacMahon et al., 2005). There are several CAD systems for automatic lung nodule detection, with some works with deep neural networks recently (Gurcan et al., 2002; Ciompi et al., 2015; van Ginneken et al., 2015; Setio et al., 2016). It is necessary to detect the lung nodules in 3D for better discrimination from vessels and reduction of false positive rate.

In our work, the solution from the second place in the 2017 National Data Science Bowl held by Kaggle was used (de Wit & Hammack, 2017). It was a patch-based solution that worked on 3D patches of size $32 \times 32 \times 32$ with $1 \times 1 \times 1$ mm$^3$ spatial resolution. Patches that contained nodules were labeled as 1 whereas the rest of the patches were labeled as 0. A coarse segmentation of lung was performed so that the detection would only perform in the lung region. The cross entropy was minimized between the predictions and the labels during the training. The structure of the detection neural network is shown in figure 3.

### 2.4 END-TO-END TRAINING OF THE ENTIRE DEEP NEURAL NETWORK

The end-to-end DNN was the joined version of the reconstruction and detection DNN proposed in section 2.2 and 2.3. Cross entropy for nodule detection was minimized for the training of the end-to-end DNN. Problem of insufficient memory of graphics processing unit (GPU) may arise because that the reconstruction DNN cannot be trained on local patches and is very memory consuming. Denote

Input 32×32×32×1 | Avg. Pool 1×1×2 | Conv 3×3×3×64 | ReLU | Max Pool 2×2×1 | Conv 3×3×3×128 | ReLU | Max Pool 2×2×2 | Dropout 0.5 | Conv 3×3×3×256 | ReLU | Max Pool 2×2×2 | Dropout 0.5 | Conv 3×3×3×512 | ReLU | Max Pool 2×2×2 | Dropout 0.5 | Conv 2×2×2×64 | ReLU | Conv 1×1×1×1 | Sigmoid | Output 1×1×1×1

Figure 3: The structure of the detection neural network. The modules in dashed boxes did not use padding, whereas the rest used zero padding if applicable.

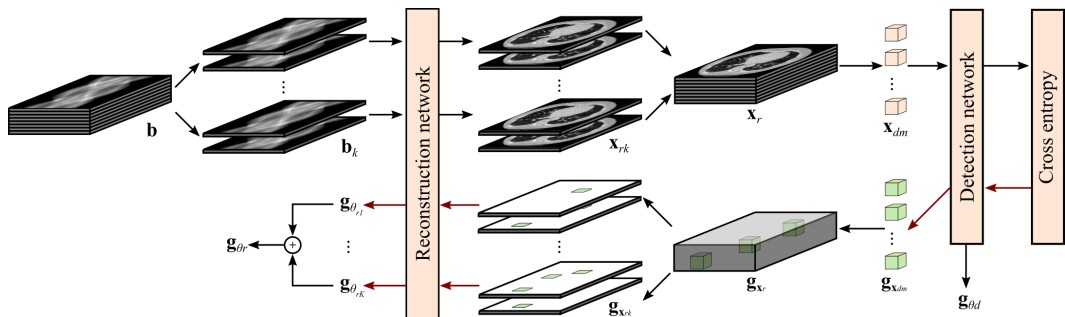

Figure 4: The schematics for joined training of the reconstruction and detection neural networks. Red arrows stand for backpropagation through neural networks. $\mathbf{g}_{\theta_{r1}}, ..., \mathbf{g}_{\theta_{rK}}$ stand for the gradients accumulated in step 16 in algorithm 1.

the reconstruction network as $g_r(\mathbf{b}; \theta_r)$ and the detection network as $g_d(\mathbf{x}; \theta_d)$, algorithm 1 was proposed to overcome this problem. And a graphical demonstration of the algorithm is presented in figure 4.

In algorithm 1, step 3 to 9 was forward propagation, and step 10 to 18 was backpropagation. The weights of the neural network were initialized from the two-step trained results. The number of layers in $\mathbf{b}$ for each iteration should be at least larger than the depth of the cubes used for detection, which was 32 in our case. The sub-layers $\mathbf{b}_k$ from $\mathbf{b}$ were not overlapped with each other to reduce computational load and complexity of the algorithm.

The patches were extracted within the lung regions in $\mathbf{x}_r$, where the masks for lung regions were pre-calculated. The patches need not to fill the entire image and can also be overlapped. Denser sampling on the lung nodules was used due to the lack of positive samples compared to negative samples. Patches were also augmented with flips before being fed to the detection network, and the same flips were applied to $\mathbf{g}_{\mathbf{x}_{dm}}$ when aggregating $\mathbf{g}_{\mathbf{x}_r}$ in step 13.

Step 11 to step 13 backpropagated the gradients from patches to the entire image. Let

$$\mathbf{x}_{dm} = \mathbf{E}_m \mathbf{x}_r, \tag{7}$$

where $\mathbf{E}_m$ was the patch extraction matrix for the $m$th patch. The gradient w.r.t. $\mathbf{x}_r$ is:

$$\nabla_{\mathbf{x}_r} l = \frac{1}{M} \sum_m \mathbf{E}_m^T \nabla_{\mathbf{x}_{dm}} l_m, \tag{8}$$

which gave step 11 to step 13 in algorithm 1.

The same analysis could be applied to the calculation of gradient backpropagation from $\mathbf{x}_r$ to parameters $\theta_r$. The reconstructed images were

$$\mathbf{x}_r = \sum_k \mathbf{E}_k^T \mathbf{x}_{rk} = \sum_k \mathbf{E}_k^T g_r(\mathbf{b_k}; \theta_r), \tag{9}$$

---

**Algorithm 1** Joined training of the reconstruction and detection networks

---

**Require:** $\theta_{r0}, \theta_{d0}$: pre-trained weights of reconstruction and detection networks
**Require:** Other necessary data and configurations for the training
1: **Initialization** Initialize $\theta_r \leftarrow \theta_{r0}, \theta_d \leftarrow \theta_{d0}$
2: **while** not converged **do**
3:     Get projection data $\mathbf{b}$ with enough successive layers
4:     **for each** $L$ sub successive layers $\mathbf{b}_k$ in $\mathbf{b}$ **do**
5:         $\mathbf{x}_{rk} \leftarrow g_r(\mathbf{b}_k; \theta_r)$
6:     **end for**
7:     Aggregate $\mathbf{x}_{rk}$ to get the reconstructed image $\mathbf{x}_r$
8:     Extract patches $\mathbf{x}_{d1}, \cdots, \mathbf{x}_{dM}$ and their corresponding labels $z_1, \cdots, z_M$ from $\mathbf{x}_r$
9:     Define $l_m = \mathtt{crossentropy}(g_d(\mathbf{x}_{dm}; \theta_d), z_m)$
10:     $\mathbf{g}_{\theta_d} \leftarrow \frac{1}{M} \nabla_{\theta_d} \sum_m l_m$ (Gradient of the detection network)
11:     $\mathbf{g}_{\mathbf{x}_{dm}} \leftarrow \frac{1}{M} \nabla_{\mathbf{x}_{dm}} l_m$ **for each** $m$ (Gradient of the patches)
12:     $\mathbf{g}_{\mathbf{x}_r} \leftarrow \mathbf{0}$ (Same dimension with $\mathbf{x}_r$)
13:     Add $\mathbf{g}_{\mathbf{x}_{dm}}$ to $\mathbf{g}_{\mathbf{x}_r}$ according to the position of $\mathbf{x}_{dm}$ **for each** $m$
14:     $\mathbf{g}_{\theta_r} \leftarrow \mathbf{0}$ (Gradient of the reconstruction network)
15:     **for each** $L$ sub successive layers $\mathbf{g}_{\mathbf{x}_{rk}}$ in $\mathbf{g}_{\mathbf{x}_r}$ **do**
16:         $\mathbf{g}_{\theta_r} \leftarrow \mathbf{g}_{\theta_r} + \mathbf{J}_{\mathbf{x}_{rk}}(\theta_r) \mathbf{g}_{\mathbf{x}_{rk}}$ (Accumulate gradients from all the sub layers)
17:     **end for**
18:     Apply $\mathbf{g}_{\theta_r}, \mathbf{g}_{\theta_d}$ to $\theta_r$ and $\theta_d$ respectively
19: **end while**
20: **return** $\theta_r, \theta_d$ network parameters for reconstruction and detection networks

---

where $\mathbf{E}_k$ was the extraction matrix for the $k$th sub-layers. There was no normalization factors because non-overlapped sub-layers were used. The gradient w.r.t. $\theta_r$ is:

$$\nabla_{\theta_r} l = \mathbf{J}_{\mathbf{x}_r}(\theta_r) \nabla_{\mathbf{x}_r} l = \sum_k \mathbf{J}_{\mathbf{x}_{rk}}(\theta_r) \mathbf{E}_k \sum_k \mathbf{E}_k^T \nabla_{\mathbf{x}_{rk}} l = \sum_k \mathbf{J}_{\mathbf{x}_{rk}}(\theta_r) \nabla_{\mathbf{x}_{rk}} l, \qquad (10)$$

which gave step 16 in algorithm 1. Because $\mathbf{E}_k$ were non-overlapped, it holds that

$$\mathbf{E}_k \mathbf{E}_l^T = \begin{cases} \mathbf{I}, & k = l \\ \mathbf{0}, & k \neq l \end{cases}, \qquad (11)$$

which canceled the $\mathbf{E}_k$ in (11). $\mathbf{J}_{\mathbf{x}}(\theta)$ is the Jacobian matrix defined as $J_{ij} = \partial x_i / \partial \theta_j$, which was obtained by backpropagation through the reconstruction neural network.

The step 18 was applying changes to the weights of the neural network according to the gradients. Adam algorithm was used in our realization.

## 3  SIMULATION SETUP

### 3.1  DATA SOURCE

The TCIA LIDC-IDRI dataset was used for the simulation. The dataset contains 1,018 chest CT scans from various scanners, which were annotated by 4 experienced thoracic radiologists. Each lesion was annotated as one of non-small nodule ($\geq$ 3mm), small nodule ($<$ 3mm) and non-nodule (lesion that $\geq$ 3mm but not nodule). All the images were resampled to spatial resolution of $1 \times 1 \times 1$ mm$^3$ and padded by zeros to $512 \times 512$ in axial planes. The pixel value of the images was converted from Hounsfield Unit (HU) to attenuation coefficient $\mu$ in $mm^{-1}$ by

$$\mu = (HU + 1000)/1000 \times 0.02. \qquad (12)$$

We split the dataset into the training set and testing set, which contained 916 and 102 images respectively.

Table 1: Geometric parameters of the imaging model

| PARAMETERS | VALUE |
|---|---|
| Views per rotation | 720 |
| Detector resolution | 736 per layer |
| Detector pixel size | $1.2858 \times 1$ mm$^2$ |
| Source-center distance | 595 mm |
| Source-detector distance | 1086.5 mm |

## 3.2 IMAGING MODEL

To simulate low-dose CT scans, the images were forward projected with multi-slice fan-beam geometry as shown in figure 2(b). The mean photon counts out of the X-ray source for each detector unit was set to $N_0 = 1000$, which corresponded to an ultra low-dose scan compared to commercial CT scanners. The detected photon counts $N_i$ were assumed following Poisson distribution with mean value

$$\overline{N}_i = N_0 \exp(-\int_i \mathbf{x}d\mathbf{l}), \tag{13}$$

where $\int_i \mathbf{x}d\mathbf{l}$ is the line integral along the ray corresponding to the $i$th detector unit. The projection data on the $i$th detector unit is

$$b_i = \max\{-\log(N_i/N_0), b_{\mathtt{max}}\}, \tag{14}$$

where $b_{\mathtt{max}} = 10$ is to prevent overflow of $b_i$ when the attenuation is large. The geometric parameters of the imaging model is summarized in table 1.

## 3.3 RECONSTRUCTION NETWORK

The reconstruction iteration (5) was repeated for 10 times with independent $\lambda_n$ and $f_n(\cdot)$ for each iteration. The input to the reconstruction neural network was 5 successive layers, corresponding to $L = 5$ in algorithm 1. Adam algorithm with step size of $1 \times 10^{-4}$ was used for the training (Kingma & Ba, 2014). All the weights were initialized following Gaussian distribution (Tensorflow's default initializer) except that the PReLU units were initialized to ReLU (Abadi et al., 2016). For each iteration 5 random successive layers were used, and 5,500 iterations were run in total for the training of the reconstruction neural network.

## 3.4 DETECTION NETWORK

All the data were reconstructed with the trained reconstruction network for the training of the nodule detector. We only considered non-small nodules for the detection. For each annotation on non-small nodules, a positive sample was extracted at the center of the annotation and augmented by 10 times by applying randomly flips and translation near the center. Each non-small nodule was annotated 3 to 4 times in total by different radiologist, and 30 to 40 positive samples were generated for each non-small nodule.

The negative samples were composed of 3 parts: radiologists-annotated non-nodules, tissues in lung and tissues on the edge of lung. The non-nodules were augmented by 4 times by flips and translation. For each CT image, 400 patches were randomly selected in the lung region which was segmented from the reconstruction network results, and another 100 patches were randomly selected on the edges of the same mask. All the negative patches were kept away from the center of non-small nodules with a safety margin of 64 mm.

64,720 positive samples and 522,694 negative samples were generated in total for the training of the detection network. Adam algorithm with step size of $1 \times 10^{-4}$ was used for the training. A mini-batch size of 100 was used and 12 epochs were run.

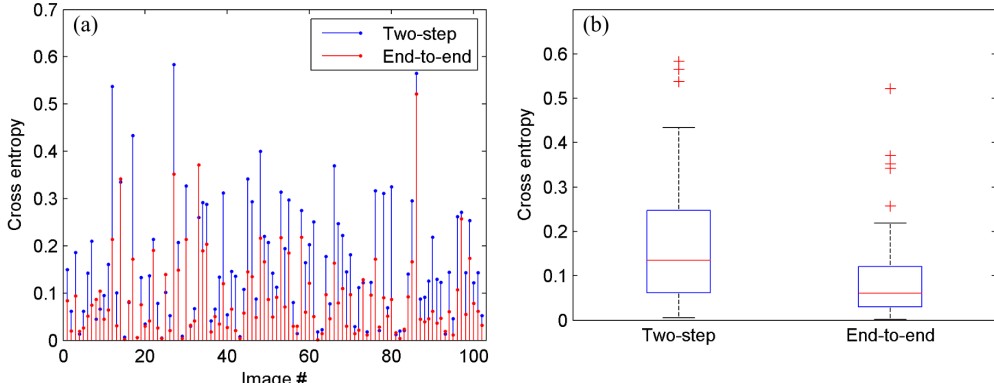

Figure 5: The patch-wise cross-entropy losses of validating CT images: (a) the cross-entropy loss of each individual image; (b) the box-and-whisker plot of the individual losses.

## 3.5 END-TO-END TRAINING

After training of the reconstruction and detection networks, a fine-tune was carried out based on trained weights according to algorithm 1. 40 successive layers were used for each iteration. To ensure a certain portion of positive samples in the training, 3 of the 6 iterations in each CT image were ensured to have non-small nodules inside the sub-images and the rest 3 were extracted randomly. 1 epoch was run for the fine-tuning step.

## 3.6 VALIDATION

Patches were extracted from the testing dataset in the same way with the training dataset. Patch-wise cross entropy loss were calculated for both two-step and end-to-end trained neural networks. ROC analysis were also performed by changing the decision threshold from 0 to 1.

It was mentioned in section 2.4 that the reconstructions were carried out on non-overlapped layers to reduce computational load. However, it was observed in our study that obvious streaking artifacts were presented in the coronal and sagittal planes, due to the fact that there was no control over the intensities of the reconstructed images in the end-to-end training. The presence of the streaking artifacts indicated that the detection network was insensitive to the different biases across slices. Thus, an overlapped reconstruction with step size of 1 were performed to eliminate the artifacts, and analyzed with the trained detection neural network.

## 3.7 IMPLEMENTATION

All the reconstruction and neural network operations were implemented on a Tesla K40m with CUDA and Tensorflow. The reconstruction operations were embedded into the Tensorflow framework via the "User Ops" interface. The patch extraction and gradient aggregation steps (step 8 and 13 in algorithm 1) were CPU based. And the gradient backpropagation step 16 of the reconstruction neural network was implemented by overriding the gradient of a dummy operation with $\mathbf{g}_{\mathbf{x}_{rk}}$.

# 4 RESULTS

## 4.1 LUNG NODULE DETECTION ACCURACY

Patch-wise entropy losses on the validation dataset were calculated for two-step and end-to-end training. The cross entropy losses were calculated for each image in the validation dataset and the results are presented in figure 5.

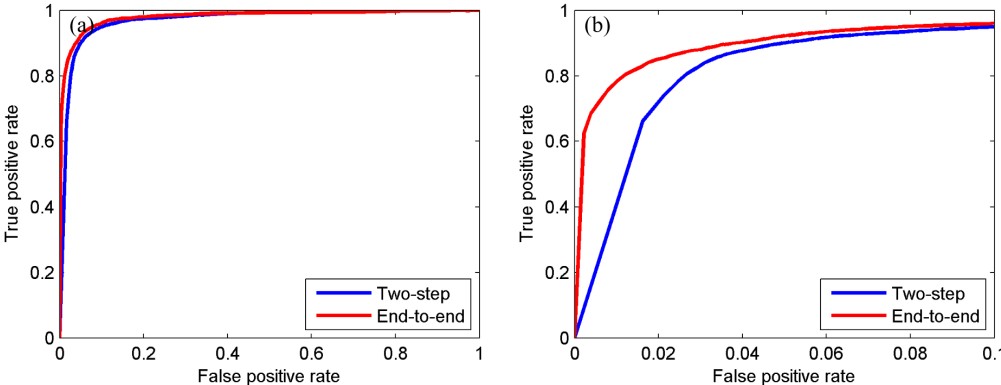

Figure 6: ROC curve of the lung nodule detection: (a) the entire ROC curve; (b) the ROC curve in the range that false positive rate is smaller than 0.1.

Table 2: Mean patch-wise cross-entropy and AUC

| METHOD | ENTROPY | AUC |
|--------|---------|-----|
| Two-step | 0.1598 | 0.9715 |
| End-to-end | 0.0904 | 0.9814 |

ROC study was also performed on the validation set with the 2 approaches, and the results are demonstrated in figure 6. The mean cross-entropy and area under curve (AUC) were calculated and summarized in table 2.

There was a significant improvement on the cross entropy with the end-to-end training scheme compared to the two-step training. According the the ROC analysis, the end-to-end method had significant gain of sensitivity of lung nodule when the false positive rate was controlled small, which was crucial for abnormality detection task.

To further analyze the source of improvement, the cross entropy was calculated for samples from 4 different sources: non-small nodules which are the positive samples, non-nodules which are suspicious lesions but not nodules, random negative samples from the lung region and random negative samples from the edge of lung. The statistical results are presented in figure 7. The biggest advantage of the end-to-end detection compared to the tow-step detection was the significantly reduced error on the non-nodules, and small gains were also presented on the cross entropy for the other 3 types of sample.

### 4.2 RECONSTRUCTED IMAGES

The outputs of the reconstruction networks for one of the validating images were plotted in figure 8. The DNN based reconstruction results had greatly reduced noise in the lung region compared to FBP, which is a standard reconstruction algorithm in clinical scanners. The results from the end-to-end training still resemble standard CT images, but with better contrast and more details in the lung region compared to the tow-step results, which have smoother appearance and lower noise level.

Figure 9 demonstrated a few examples with correct classification from one of the two methods, but incorrect classification from the other one. The end-to-end network reconstructed more details, which helped in discrimination of suspicious lesions, as shown in figure 9(a) and 9(b). However, as demonstrated in figure 9(d), the increased noise on the background may also suppress the nodule's presence.

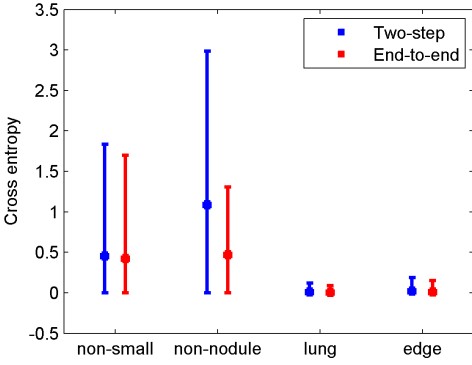

Figure 7: Cross entropy losses of different types of sample. The error bar is the standard deviation of the losses, which was truncated to zeros at the lower bounds.

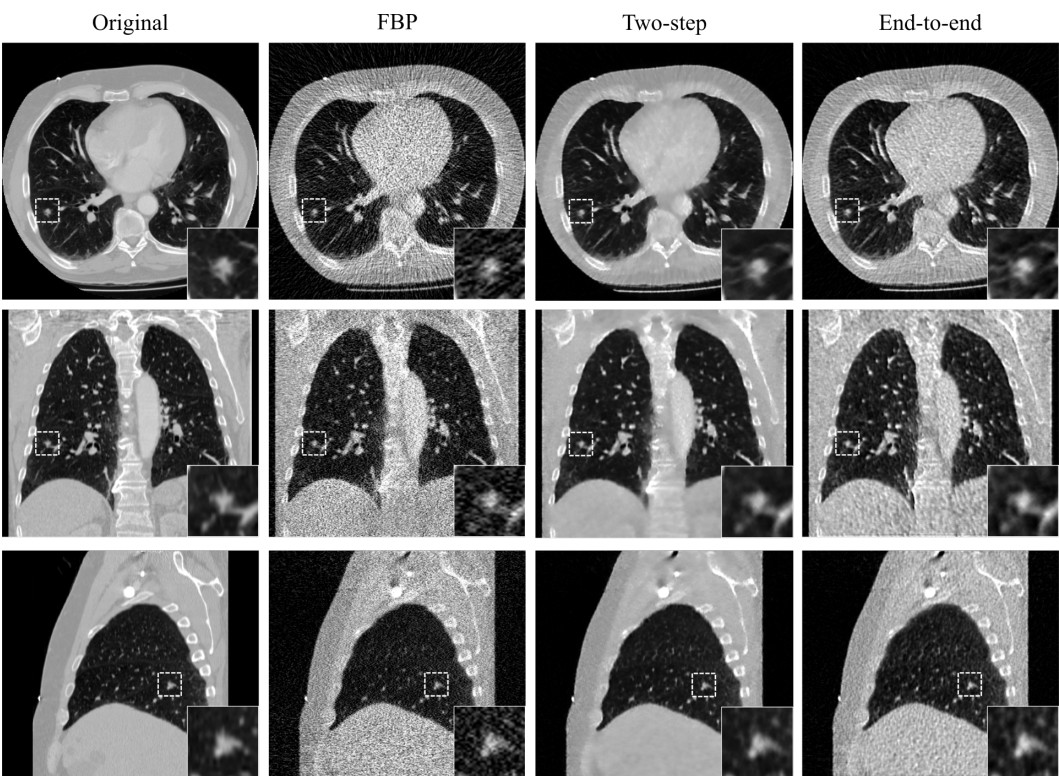

Figure 8: The reconstructed CT images for one of the validating images. From top to bottom are axial, coronal and sagittal views respectively. The FBP images were reconstructed with Hann filter, which was designed for noise suppresion. The lung nodule was annnotated with white dashed boxes and zoomed in at the lower right in each view. The display window was [-1000HU, 500HU] for all the images.

### 4.3 LEARNED FEATURES

Figure 10 demonstrated the images at different number of iterations in the reconstruction network. The images had similar appearance to ordinary CT images, except the presence of small biases in early iterations. The noise was also gradually removed from the images as it is in normal iterative reconstruction algorithms.

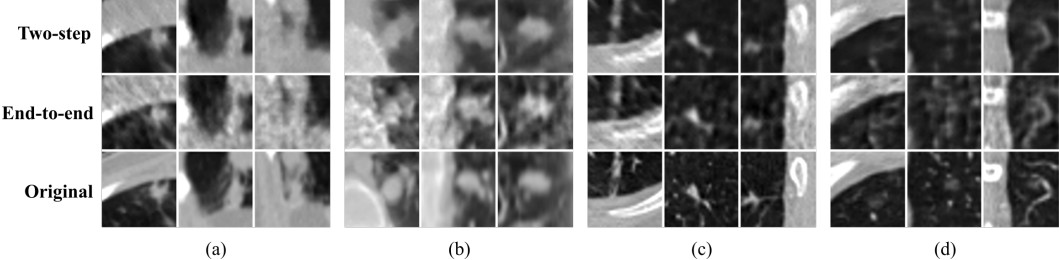

Figure 9: Example nodules: the predictions and labels of (a) - (d) are shown in table 3. For each sub-figure, from left to right were the axial, coronal and sagittal views of the suspicious lesion. The display window was [-1000HU, 500HU].

Table 3: Predictions and labels for nodules in figure 9

| METHOD | (a) | (b) | (c) | (d) |
|--------|-----|-----|-----|-----|
| Two-step | 0.97681 | 0.03627 | 0.045245 | 0.99454 |
| End-to-end | 0.12853 | 0.96602 | 0.96623 | 0.00011 |
| Label | 0 | 1 | 0 | 1 |

Figure 11 plotted some of the CNN feature maps during reconstruction, which could give some insights into the difference of the image reconstruction step between the two-step and end-to-end networks. Correlations between the feature maps were calculated for each channel, and the most correlated and uncorrelated feature maps were presented figure 11. As observed in figure 11, the most correlated components were the structures of the images, whereas the most deviated part was high-frequency components such as edges. The tissues in lung had much better contrast in the end-to-end network compared to that in the two-step network, which indicated that the end-to-end training emphasized on the lung tissue preservation.

## 4.4 CONCLUSION AND DISCUSSION

In this paper a novel end-to-end DNN was proposed for abnormality detection in medical imaging. A reconstruction network and detection network were trained jointly to maximize the abnormality detection accuracy. We implemented the method on simulated chest CT data and achieved higher non-small lung nodule detection accuracy compared to two-step training scheme. There was significant false positive rate reduction on suspicious lesions (annotated non-nodules), and fair improvement on the overall detection sensitivity and accuracy. The images reconstructed by the end-to-end method resembled ordinary CT images, with more details and increased noise level compared to that from a two-step approach.

Among the 102 validation cases, the mean entropy loss of nodule detection of the end-to-end method was smaller or similar than the two-step method for most of the cases, which indicated a statistical improvement on nodule detection from the proposed method. However, there was one case where the end-to-end entropy loss was significantly higher than the two-step loss. We studied the case further and confirmed that it was due to a strong misclassification of the positive samples, which was shown in figure 9(d).

Although there was no significant improvement on the total AUC as shown in table 2, the ROC study in figure 6 indicated a significantly improved true positive rate at small false positive rate. The U.S. carried a national lung cancer screening with low-dose CT, which was considered to cause over-diagnosis due to the high false positive rate (Team et al., 2011; Patz et al., 2014). The sensitivity improvement on the low false positive rate end indicated that the end-to-end DNN had great potential value to cancer screening tasks.

There was great difference in the appearance of the reconstructed images from the two methods. The two-step training gave images with smaller overall noise level, but some of the details in lung

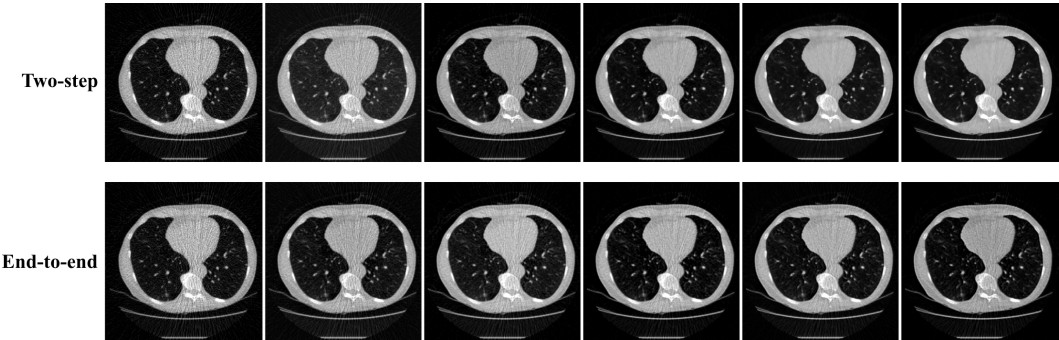

Figure 10: The recosntructed images at different iterations in the reconstruction network. From the left to right columns, the images are initial FBP, results after 2, 4, 6, 8, 10 iterations. The display window is [-1000HU, 500HU].

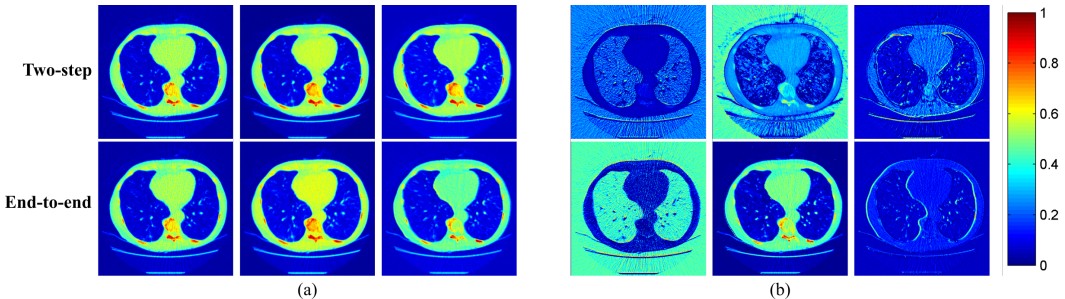

Figure 11: Feature maps from intermediate convoltuional layers of the reconstruction neural network (after applying pReLU): (a) the feature maps with the highest correlation from the same channel in two-step and end-to-end networks. (b) the feature maps with the lowest correlation from the same channel in the two networks. The feature maps were from first PReLU of iteration 1, first PReLU of iteration 5, and second PReLU of iteration 10, respectively from the left to right columns. All the feature maps were normalized to [0,1] independently.

were smoothed out, which caused misclassification for the detection network, as shown in figure 9(a) and (b). The end-to-end training revealed more details in lung with higher spatial resolution, and the images were more suitable for the automatic nodule detection task. Though there were some misclassification due to the increased noise level, the overall performance of the nodule detection was improved with the end-to-end training scheme.

The analysis on the intermediate results from the reconstruction network further revealed the difference between the two approaches. Whereas both methods kept similar structural component, the end-to-end method had more focus on the edges and tissues inside lung compared to the two-step method. As observed in figure 11(b), the structures of the lung tissue were much more clearer in the end-to-end networks. This observation indicated that sharper edge and structures were of more importance for the detection network than the noise level in the reconstructed images, which is in accordance with human perceptions when radiologists perform the same task.

Selecting appropriate representations of the data for further tasks such as detection is crucial, and the philosophy in end-to-end training is to leave the representation selection problem to machine rather than hand-crafting it. In this work, we demonstrated the feasibility of the end-to-end DNN for abnormality detection in medical imaging for a specific lung nodule detection problem in chest CT, and concluded that better results can be achieved for CAD systems by doing so. Nowadays most CAD systems are trained on reconstructed images, which are designed and tuned for radiologists rather than machines. By integrating the reconstruction process into the detection pipeline, better detection accuracy could be achieved for the CAD systems, and will increase its value to radiologists.

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
