# OpenReview forum: "End-to-End Abnormality Detection in Medical Imaging"
_ICLR.cc/2018/Conference — Reject_

### Official Review · AnonReviewer3 · 2017-11-28
**A well written paper showing the promise of DNNs for solving tough inverse imaging problems. The contributions seem incremental, not properly enunciated, or appropriately validated.**

**Rating:** 4
**Confidence:** 4

**Review:**

The paper proposes a DNN for patch-based lung nodule detection, directly from the CT projection data. The two-component network, comprising of the reconstruction network and the nodule detection network, is trained end-to-end. The trained network was validated on a simulated dataset of 1018	low-dose chest CT images. It is shown that end-to-end training produces better results compared to a two-step approach, where the reconstruction DNN was trained first and the detection DNN was trained on the reconstructed images.

Pros

It is a well written paper on a very important problem. It shows the promise of DNNs for solving difficult inverse problems of great importance. It shows encouraging results as well.

Cons

The contributions seem incremental, not properly enunciated, or appropriately validated.

The putative contributions of the paper can be
(a) Directly solving the target problem from raw sensory data without first solving an inversion problem
(b) (Directly) solving the lung nodule detection problem using a DNN.
(c) A novel reconstruction DNN as a component of the above pipeline.
(d) A novel detection network as a component of the above pipeline.

Let's take them one by one:

(a) As pointed out by authors, this is in the line of work being done in speech recognition, self-driving cars, OCR etc. and is a good motivation for the work but not a contribution. It's application to this problem can require significant innovation which is not the case as components have been explored before and there is no particular innovation involved in using them together in a pipeline either.

(c) As also pointed by the authors, there are many previous approaches - Adler & Oktem (2017), Hammernik et al (2017) etc. among others. Another notable reference (not cited) is Jin et al. "Deep Convolutional Neural Network for Inverse Problems in Imaging." arXiv preprint arXiv:1611.03679 (2016). These last two (and perhaps others) train DNNs to learn unrolled iterative methods to reconstruct the CT image. The approach proposed in the paper is not compared by them (and perhaps others), neither at a conceptual level nor experimentally. So, this clearly is not the main contribution of the paper.

(d) Similarly, there is nothing particularly novel about the detection network nor the way it is used.

This brings us to (b). The proposed approach to solve this problem may indeed by novel (I am not an expert in this application area.), but considering that there is a considerable body of work on this problem, the paper provides not comparative evaluation of the proposed approach to published ones in the literature. It just provides an internal comparison of end-to-end training vis-a-vis two step training.

To summarize, the contributions seem incremental, not properly enunciated, or appropriately validated.

---

### Official Review · AnonReviewer2 · 2017-11-28
**Practical work in a useful direction in medical image analysis**

**Rating:** 5
**Confidence:** 4

**Review:**

This paper proposes to jointly model computed tomography reconstruction and lesion detection in the lung, training the mapping from raw sinogram to detection outputs in an end-to-end manner. In practice, such a mapping is computed separately, without regard to the task for wich the data is to be used. Because such a mapping loses information, optimizing such a mapping jointly with the task should preserve more information that is relevant to the task. Thus, using raw medical image data should be useful for lesion detection in CT as well as most other medical image analysis tasks.


Style considerations:

The work is adequately motivated and the writing is generally clear. However, some phrases are awkward and unclear and there are occasional minor grammar errors. It would be useful to ask a native English speaker to polish these up, if possible. Also, there are numerous typos that could nonetheless be easily remedied with some final proofreading. Generally, the work is well articulated with sound structure but needs polish.

A few other minor style points to address:
- "g" is used throughout the paper for two different networks and also to define gradients - if would be more clear if you would choose other letters.
- S3.3, p. 7 : reusing term "iteration"; clarify
- fig 10: label the columns in the figure, not in the description
- fig 11: label the columns in the figure with iterations
- fig 8 not referenced in text


Questions:

1. Before fine-tuning, were the reconstruction and detection networks trained end-to-end (with both L2 loss and cross-entropy loss) or were they trained separately and then joined during fine-tuning?
(If it is the former and not the latter, please make that more clear in the text. I expect that it was indeed the former; in case that it was not, I would expect fully end-to-end training in the revision.)

2. Please confirm: during the fine-tuning phase of training, did you use only the cross-entropy loss and not the L2 loss?

3a. From equation 3 to equation 4 (on an iteration of reconstruction), the network g() was dropped. It appears to replace the diagonal of a Hessian (of R) which is probably a conditioning term. Have you tried training a g() network? Please discuss the ramifications of removing this term.

3b. Have you tracked the condition number of the Jacobian of f() across iterations? This should be like tracking the condition number of the Hessian of R(x).

4. Please discuss: is it better to replace operations on R() with neural networks rather than to replace R()? Why?

5. On page 5, you write "masks for lung regions were pre-calculated". Were these masks manual segmentations or created with an automated method?

6. Why was detection only targetted on "non-small nodules"? Have you tried detecting small nodules?

7. On page 11, you state: "The tissues in lung had much better contrast in the end-to-end network compared to that in the two-step network". I don't see evidence to support that claim. Could you demonstrate that?

8. On page 12, relating to figure 11, you state:

"Whereas both methods kept similar structural component, the end-to-end method had more focus on the edges and tissues inside lung compared to the two-step method. As observed in figure 11(b), the structures of the lung tissue were much more clearer in the end-to-end networks. This observation indicated that sharper edge and structures were of more importance for the detection network than the noise level in the reconstructed images, which is in accordance with human perceptions when radiologists perform the same task."

However, while these claims appear intuitive and such results may be expected, they are not backed up by figure 11. Looking at the feature map samples in this figure, I could not identify whether they came from different populations. I do not see the evidence for "more focus on the edges and tissues inside lung" for the end-to-end method in fig 11. It is also not obvious whether indeed "the structures of the lung tissue were much more clearer" for the end-to-end method, in fig 11. Can you clarify the evidence in support of these claims?


Other points to address:

1. Please report statistical significance for your results (eg. in fig 5b, in the text, etc.). Also, please include confidence intervals in table 2.

2. Although cross-entropy values, detection metrics were not (except for the ROC curve with false positives and false negatives). Please compute: accuracy, precision, and recall to more clearly evaluate detection performance.

3a. "Abnormality detection" implies the detection of anything that is unusual in the data. The method you present targets a very specific abnormality (lesions). I would suggest changing "abnormality detection" to "lesion detection".

3b. The title should also be updated accordingly. Considering also that the presented work is on a single task (lesion detection) and a single medical imaging modality (CT), the current title appears overly broad. I would suggest changing it from "End-to-End Abnormality Detection in Medical Imaging" -- possibly to something like "End-to-End Computed Tomography for Lesion Detection".


Conclusion:

The motivation of this work is valid and deserves attention. The implementation details for modeling reconstruction are also valuable. It is interesting to see improvement in lesion detection when training end-to-end from raw sinogram data.  However, while lung lesion detection is the only task on which the utility of this method is evaluated, detection improvement appears modest. This work would benefit from additional experimental results or improved analysis and discussion.

---

### Official Review · AnonReviewer1 · 2017-12-05
**Interesting direction, but let's check the details**

**Rating:** 6
**Confidence:** 3

**Review:**

The authors present an end to end training of a CNN architecture that combines CT image signal processing and image analysis. This is an interesting paper. Time will tell whether a disease specific signal processing will be the future of medical image analysis, but - to the best of my knowledge - this is one of the first attempts to do this in CT image analysis, a field that is of significance both to researchers dealing with image reconstruction (denoising, etc.) and image analysis (lesion detection).  As such I would be positive about the topic of the paper and the overall innovation it promises both in image acquisition and image processing, although I would share the technical concerns pointed out by Reviewer2, and the authors would need good answers to them before this study would be ready to be presented.

---

### Decision · Program_Chairs · 2018-01-29
**ICLR 2018 Conference Acceptance Decision**

**Decision:**

Reject

**Comment:**

Authors present an evaluation of end-to-end training connecting reconstruction network with detection network for lung nodules.

Pros:
- Optimizing a mapping jointly with the task may preserve more information that is relevant to the task.

Cons:
- Reconstruction network is not "needed" to generate an image -- other algorithms exist for reconstructing images from raw data. Therefore, adding the reconstruction network serves to essentially add more parameters to the neural network. As a baseline, authors should compare to a detection-only framework with a comparable number of parameters to the end-to-end system. Since this is not provided, the true benefit of end-to-end training cannot be assessed.

- Performance improvement presented is negligible

- Novelty is not clear / significant